# Patient Perceptions of Electronic Prescriptions in Belgium: An Exploratory Policy Analysis

**DOI:** 10.3390/pharmacy6040130

**Published:** 2018-12-08

**Authors:** Laura Suykerbuyk, Marieke Robbrecht, Simon De Belder, Hilde Bastiaens, Wim Martinet, Hans De Loof

**Affiliations:** 1Laboratory of Physiopharmacology, University of Antwerp, Universiteitsplein 1, B-2610 Antwerp, Belgium; ls@sensio.be (L.S.); wim.martinet@uantwerpen.be (W.M.); 2Department of Primary and Interdisciplinary Care, Faculty of Medicine and Health Sciences, University of Antwerp, Universiteitsplein 1, B-2610 Antwerp, Belgium; mariekerobbrecht@gmail.com (M.R.); debelder.s@hotmail.com (S.D.B.); hilde.bastiaens@uantwerpen.be (H.B.)

**Keywords:** electronic prescribing, statistics and numerical data, community pharmacy services, organization and administration, public opinion, surveys and questionnaires, health services administration

## Abstract

In today’s world, digitalization and automation are ubiquitous and different countries have different strategies for implementing information technology in health care. In Belgium, these plans include the dematerialization of prescriptions, following the implementation of a system of electronic prescribing. In the light of these changes, we studied the attitudes of patients toward a paperless prescription. We collected the opinions of 273 patients by survey. Older people, or people with more complex medical needs, expressed a clear desire to keep receiving a paper version of the prescription. Younger people foresaw practical advantages, and expressed a willingness to buy prescription-only medicines online. Knowledge about the planned changes was, however, limited. Privacy and autonomy issues were expressed by a large fraction of people. The problem of what happens when a third person wants to pick up medicines for a patient, a frequent occurrence, was often mentioned. We conclude that, at present, patients have inadequate knowledge and understanding about the planned changes. In light of these considerations and the recent technical problems encountered by the system, we feel that an updated risk/benefit analysis of the planned policy is urgently needed.

## 1. Introduction

Digitalization and automation are ubiquitous in our society [1], and this is no different in the health care sector, where big benefits may potentially be obtained [2,3]. Big advantages are obvious in the speed, reliability, quantity, integration, and analysis of information. These transformations are also affecting the way prescriptions are handled. It is important to realize that there is more than one way of using computer and network technology in the implementation of electronic prescribing, resulting in a broad spectrum of possible risks and benefits [4,5]. The possible benefits are a reduction of prescribing and medication errors, an increase in efficiency, and a limitation of fraud. On the negative side, legal issues, privacy concerns, implementation costs, and technical problems were noted.

The European Commission has long-term plans for interoperability between countries [6,7], but, at present, electronic prescribing is unevenly implemented in different countries [8,9,10]. Scandinavian countries are the forerunners [11], and Belgium [12] is probably a middle-ranking country.

In Belgium, at present a large number of prescriptions written by primary care physicians are made electronically, and temporarily stored on a centralized server [13]. Patients receive a printout, on which there is a code that allows authorized pharmacists to retrieve the electronic prescription (e-prescriptions) from the server. This code is, for convenience, also printed as a machine-readable bar-code. From a legal standpoint, this piece of paper is only a pointer to the “real”, legally binding e-prescription, but it is readable by the patient and useful for the pharmacist when the central server is not available—a frequent occurrence since the startup of the system [14,15,16].

Belgium has put forward plans to make e-prescriptions paperless in the near future [17,18]. The paper-based “pointer to” is to be phased out, resulting in a “paperless e-prescription”. It is obvious that a different retrieval mechanism would then be needed by the pharmacist to obtain the e-prescription. In addition, patients would only have ‘a view’ of their e-prescriptions through computerized electronic means. As of the writing of this paper, the details about the modalities for patients and pharmacists to be able to see the e-prescriptions are not available, although the decision of going paperless appears to be a fait-accompli [19].

Some legislative initiatives in other countries about e-prescriptions have recently been criticized, revealing practical problems, a failed implementation attempt, or the creation of additional barriers for vulnerable patients [20,21,22]. To the best of our knowledge, there are no other studies documenting the opinions of patients about future changes concerning e-prescriptions. Other studies documenting patient perspectives and preferences on existing systems are very rare [23]. As these innovations and legislative mandates are potentially disruptive, a study was initiated to document patients’ opinions and knowledge about the proposed transition to paperless e-prescriptions in Belgium.

## 2. Materials and Methods

To probe this question, we developed a questionnaire with 26 multiple-choice questions. Different topics were covered: Demographics, the presence of chronic medication use (self-reported), who picks up the medicines at the pharmacy, personal knowledge and experience with e-prescriptions, privacy and autonomy concerns, willingness to go paperless, and attitudes towards buying medicines online. This questionnaire was thoroughly reviewed within our team. An online version was made using the Qualtrics software (April 2018, Qualtrics, Provo, UT, USA), and a convenience sample was obtained by distributing this link electronically on social media. To reach a broader age range, a number of patient organizations were contacted, and four of these were willing to further distribute the link on their websites: ReumaNet, MS-Liga Vlaanderen, Het Ventiel (campaigns for people with early onset dementia), and Sensoa (a promoter of sexual health). In addition, a number of face-to-face interviews were initiated with seniors, to broaden the age range and reduce the bias of inquiring only through digital means about this digital topic. We did not carry out an exhaustive and detailed statistical analysis or analyze subgroups, because of the nature of the sample and its limited size. Due to the online distribution and strict anonymous nature of our survey, the theoretical possibility of duplicate entries cannot be completely excluded. This also precluded an estimation of the response rate. Questionnaires were conducted between 19 April and 27 May 2018. The full questionnaire can be found in the Appendix A. This study was deemed exempt for review by an Institutional Review Board.

## 3. Results

Figure 1 shows the age and gender distribution of the 273 respondents to our questionnaire. In addition, we show within these age groups whether medication use was chronic or acute (self-reported). 

As expected, the fraction of people taking medication on a chronic basis was clearly larger for people aged 50 or older [24], and, although more women participated in the survey, there were no marked gender differences. A total of 63% of the people reported going to the pharmacy to pick up medicines for other people, nearly always close family members. 

A clear majority of people (68–81%) declared knowing what e-prescriptions were (Figure 2a). There was no clear age-related effect. In total, 34% of the people claimed that their doctor did not write prescriptions electronically, an unrealistically high number, as >80% of primary care physicians do write e-prescriptions [25]. As people still receive a piece of paper, they may not fully realize to which extent digitalization has progressed. Here, also, there was no clear age-related bias present (Figure 2b).

A total of 37% of the respondents expressed a positive opinion about e-prescriptions, and cited readability as a primary reason. In contrast, 7% of respondents had a negative opinion, citing the faster expiration date as an important reason (in Belgium, e-prescriptions do not expire faster than hand-written prescriptions, but the system enforces the presence and readability of an expiration date).

When given the choice, 51% of respondents preferred to receive a paper printout of their prescription. This preference for a printed version was much more pronounced amongst people using medication on a chronic basis, versus those using medicines only on an acute basis (60% vs. 41%). Positive aspects mentioned regarding paperless prescriptions were primarily related to the lower ecological burden, but other advantages were also mentioned: efficiency, reduced possibility of fraud, and less visits to the doctor. Negative aspects mentioned were primarily related to the lack of ability to control of whether one still had a valid prescription, but also a lack of control about other characteristics of the medication (e.g., brand preference or galenic formulation). People also expressed the need to have a paper prescription, as a reminder to actually go and pick up the medication from the pharmacy. Another big concern expressed was the difficulty of having someone else pick up the medicines from the pharmacy, in the case of a paperless prescription.

A total of 64% of people expressed the opinion that there should be more easily accessible information available on this topic. In Figure 3, we summarize some of the other opinions expressed about paperless prescriptions.

It is clear that, for a substantial proportion of respondents (but not the majority), there were qualms about autonomy, privacy, and about keeping an overview of things, upon the introduction of paperless e-prescriptions.

Only 15% of people knew about the existence of the “Personal Health Viewer” [26], the web-portal recently introduced by the government that lets patients look at, among other things, their prescriptions. In addition, 59% thought it might be sufficient to keep track of their prescriptions, but 10% found this unacceptable, because they either did not possess or did not know how to work with a computer. In contrast, 32% expressed the need to be able to see prescriptions on their smartphone. A high percentage, 89%, thought it should always be possible for a patient to request a printed version, and 21% thought it should always be mandatory. A total of 80% were of the opinion that it is the physician who is responsible for giving a printed version to the patient.

In Belgium, prescription-only medicines cannot be sold through the internet, but because a paperless prescription could facilitate this by obviating a physical transfer of a paper prescription to the pharmacy, we asked whether people might use this option if possible. Figure 4 clearly shows that younger people (a majority of 57% in the 25–45 age group) were more open to this possibility, in contrast to the older generations. 

## 4. Discussion

We initiated this survey because we could not find any studies probing patient preferences and opinions about e-prescriptions, and the consequences of the planned implementation changes. One of the most noticeable results of this survey is the need for more information, as expressed by the majority of patients. It is also clear that they have only partial knowledge of the consequences of the planned changes. One can safely state that a substantial number of people do not realize fully that, at present, their prescriptions are already (in a large part) electronic, because they still receive a paper printout. Not surprisingly, there are concerns about privacy and autonomy, although we did not question these concerns in detail. This is a highly technical matter, beyond most people’s expertise, but the perceived lack of information most likely has a big impact on this level of uncertainty.

Another important outcome of this survey is the preponderance of people who picked up medicines in the pharmacy for someone else. At present, this task can easily and securely be delegated by handing over a paper prescription, or the pointer to the e-prescription (which is still paper-based). A substantial majority realized that this will no longer be possible with paperless implementation. As of the writing of this paper, there is no proposal for a solution to this problem from the Belgian health authorities, in spite of the stated intention of going paperless in the near future. Without a formal system documenting authorization by a patient, it is clear that the pharmacy personnel cannot possibly verify if a person is indeed trusted/authorized by the patient. A comparable situation in pharmacies in the Netherlands has already resulted in extra paperwork [27]. Without a paper prescription, these extra layers of protection are clearly necessary to keep the medical information that is intrinsically linked to the use of certain medicines (or combinations thereof) confidential. A simple example can illustrate how easily things can be jeopardized. The survey showed that partners often pick up medicines for each other, but, during a divorce, ex-partners may well wish to keep their medication use private. These conflicts may not be known in the pharmacy, and if prescriptions can be retrieved from the central server just by looking up the name of the patient, at the demand of an ex-partner, confidentiality would quickly be breached. Beyond medical confidentiality, there is ample reason to believe that current European law, including the recently introduced the General Data Protection Regulation (GPDR) [28], also necessitates this strict documentation of trust before dispensing medicine to a third person. With this in mind, it becomes very difficult to justify the advantages of going paperless, because of the addition of a new layer of bureaucracy. It is obvious that most patients, a substantial number of health care workers, and some people in charge of policy do not realize the consequences of certain technology-driven policy decisions. This is, however, not a novel situation [20,21].

When asked about the positive aspects of paperless prescriptions, younger people discerned some advantages, such as the convenience of possibly being able to obtain a prescription without the need for a physical appointment and/or the possibility of obtaining repeat prescriptions. This younger age group was also interested in obtaining their prescription-only medication online, something that is, at present, not possible in Belgium. As people age, or as people have more complex medication needs, we saw a clear preference for keeping track of things through a paper prescription.

The practical consequences of the implementation and the legal mandates surrounding e-prescriptions are complex, and success is not guaranteed, as failures have been described [22]. There are also difficult issues about how technology enables changes in practice, and how resistance to change or perceived new risks can impede implementation [29]. The system is indeed complex, and during the writing of this discussion, the system in Belgium experienced (once more) a nearly complete breakdown, for more than a day [30,31]. It hardly needs saying that, if the system had already gone paperless, the practical consequences would have been even more substantial and would constitute a definite public health problem. It is, therefore, not surprising that the date for enforcing physicians to only write e-prescriptions, and the transition to a paperless system, has been postponed on several occasions [18,32].

Possible limitations to this study were the internet-based nature of this study and the relatively small sample size, precluding exact quantitative estimates and cross-correlations of the different opinions, although it certainly was big enough to highlight important problems and hesitations. Sampling bias may still be present, although we tried to include a number of people that would ordinarily not participate in online questionnaires. Therefore, caution is needed, for example in the interpretation of the exact number of people that would not be able to personally access their paperless e-prescriptions.

We would suggest that future policy changes concerning e-prescriptions be preceded by a risk/benefit analysis, carried out by people without a conflict of interest and thus not directly involved in the running of the system. In addition, much more information for the general public should be available, in order to make changes transparent and comprehensible to everybody involved. This information should probably also be channeled through the health care workers themselves. We also realized during informal discussions about this study, that a substantial number of health care workers themselves were ill-informed, and we plan to document this in future research. We cannot ignore the benefits of automation and digitalization, but blind techno-optimism [33] should not be at the expense of personal privacy or public health.

## 5. Conclusions

Older people, or people with more complex medical needs, clearly expressed the desire to continue receiving a paper version of their prescriptions. Younger people, in contrast, envisaged advantages in paperless e-prescriptions, and expressed an interest in obtaining their prescription-only medicines online. Knowledge about these planned and implemented changes was, however, very limited, and only 15% knew about the “Personal Health Viewer”. Privacy and autonomy issues were expressed frequently, and so were problems surrounding a third person picking up medicines from the pharmacy. We conclude that, at present, patients have inadequate knowledge and understanding about the planned changes. In light of these considerations and the recent technical problems encountered by the system, we feel that an updated risk/benefit analysis of the planned policy is urgently needed. Additional analyses of system risk and rewards, from the patient perspective, may be warranted.

## Figures and Tables

**Figure 1 pharmacy-06-00130-f001:**
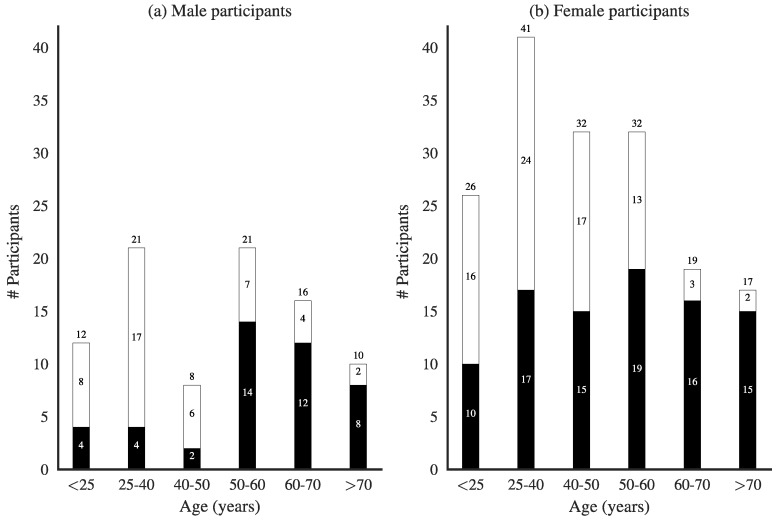
Characteristics of the respondents to the questionnaire. Age distribution of (**a**) male, and (**b**) female participants. Black bars represent self-reported chronic medication use; white bars represent acute use.

**Figure 2 pharmacy-06-00130-f002:**
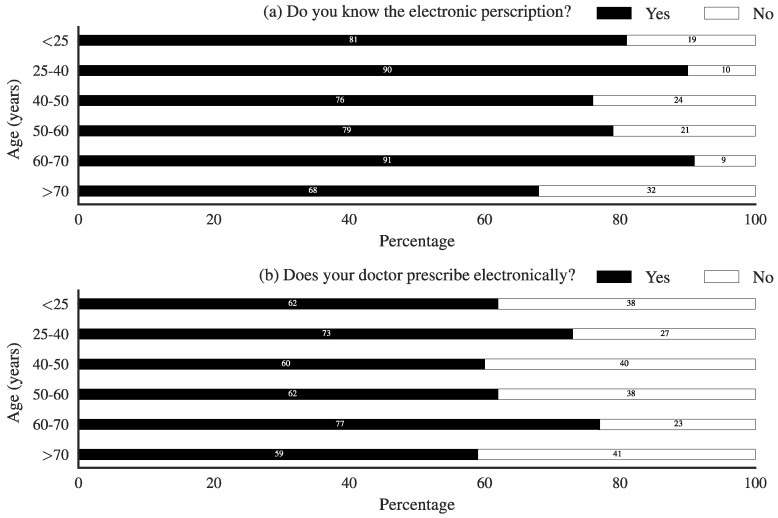
Knowledge of e-prescriptions. (**a**) Self-reported knowledge of e-prescriptions as a function of age; (**b**) Reported use by their physician of e-prescriptions in function of age.

**Figure 3 pharmacy-06-00130-f003:**
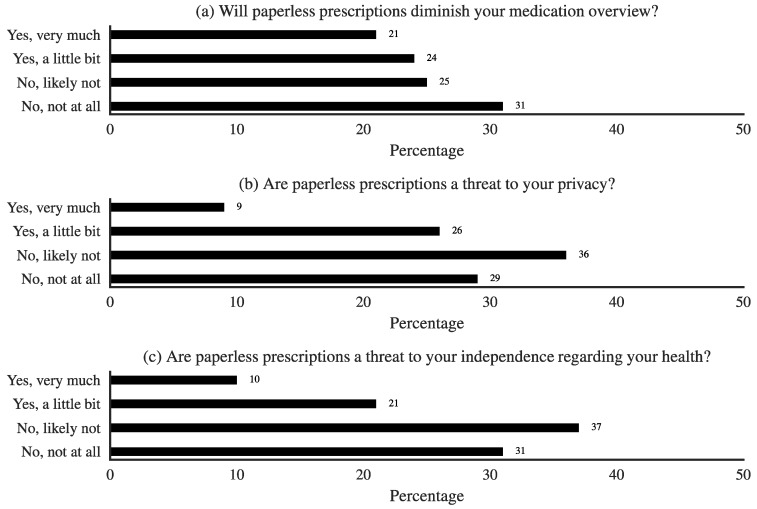
Opinions about e-prescriptions. From top to bottom: Opinions about overview, privacy, and autonomy.

**Figure 4 pharmacy-06-00130-f004:**
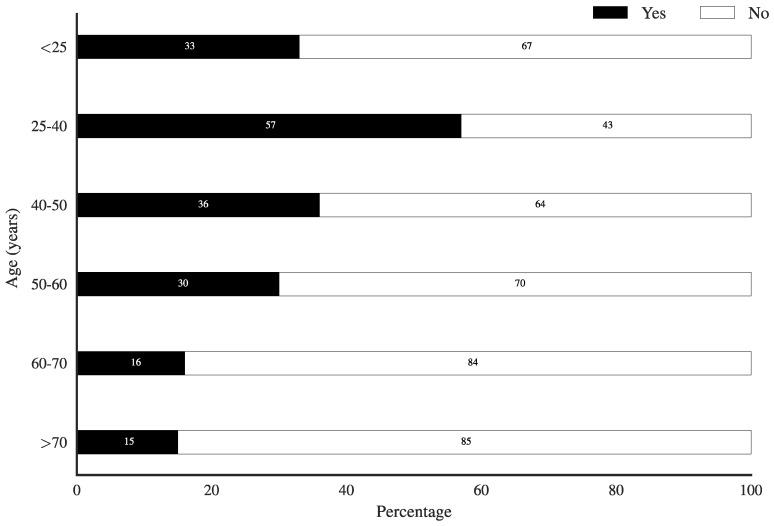
Age distribution of willingness to order prescription-only medicines online.

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
