# Peer review of "Patient Perceptions of Electronic Prescriptions in Belgium: An Exploratory Policy Analysis"

_pharmacy, 2018, doi:10.3390/pharmacy6040130_

Round 1

Reviewer 1 Report

- Almost all references are used without providing or describing the summary findings of the references. Readers will have to go to the references to understand what the authors meant to say, which is not recommended. - Certain terms are not used consistently, which may lead to confusion. For example, the authors used e-prescribing, e-prescription, and electronic prescription inconsistently in the text. Also, the authors used self-declared and self-reported, which meant to say the same. The paper-printout copy of the prescription was referred as a prescription in the introduction. The author needs to clearly state the difference between paperless electronic prescription and electronic prescription with paper-printout. These terms need to be consistent, so that the reader can follow the content. - The manuscript needs a review by a copy editor or someone who can evaluate the grammar, flow, and style of English. There were a number of issues with punctuation and typos. - It is unclear why buying medicines online was evaluated. Needs further clarification. - In terms of methods, it says the online version of the survey was distributed, but unclear to whom. Also, the link to the survey question was distributed in the website, but not sent to patients? Who was selected for face-to-face interview? what criteria was there to select these participants? There is no response rate reported either. The methods section needs to be more clear. Maybe inclusion and exclusion criteria may help. - Needs to define what was considered "chronic" and "acute" in terms of prescription use. - The results section used the words, such as "majority" and "nearly always". Needs to report percentage for these. - In line 82, "50+ people" needs to be changed to "people with age of 50 or older" - Needs a description on "Personal Health Viewer" - It is unclear how paperless electronic prescription can lead to online internet sale. - In line 160-161, the abbreviation GPDR is used, but unclear where this came from. Overall, the manuscript needs significant improvement. Also, the significance of study finding is questionable as the implementation of paperless electronic prescription is happening, regardless of this. The potential risk of this implementation is not described adequately.

Author Response

-Almost all references are used without providing or describing the summary findings of the references. Readers will have to go to the references to understand what the authors meant to say, which is not recommended. ee

We provided some extra detail in the text but it is difficult to gauge the amount of detail the reviewer needs or finds desirable. References are there to be looked up for anyone who needs more detail and we took great care in selecting them.  Elaborating on all of them would make the text much longer at the risk of becoming tedious for the majority of readers. 

 Certain terms are not used consistently, which may lead to confusion. For example, the authors used e-prescribing, e-prescription, and electronic prescription inconsistently in the text. Also, the authors used self-declared and self-reported, which meant to say the same.

We agree with the reviewer and edited these terms for consistency in the whole manuscript.

 -The paper-printout copy of the prescription was referred as a prescription in the introduction. The author needs to clearly state the difference between paperless electronic prescription and electronic prescription with paper-printout. These terms need to be consistent, so that the reader can follow the content.

These differences are explicitly explained in the introduction (2e and 3e paragraph) but we increased consistency and that should augment readability.

- The manuscript needs a review by a copy editor or someone who can evaluate the grammar, flow, and style of English. There were a number of issues with punctuation and typos.

Without clear reference to specific places in the manuscript we were unable to find “a number of issues with punctuation and typos”.  Our manuscript was evaluated and edited by a native English-speaking scientist.  Due to the vagueness of these comments they are not very helpful and we respectfully disagree as to need for a major overhaul of the language used in order to communicate our science but we have tried to improve the readability of the manuscript.  We note that the other review found the results and discussion to be “well-written”.

- It is unclear why buying medicines online was evaluated. Needs further clarification.

We clearly state the reason for this in the penultimate sentence of the results section but we expanded this sentence.

- In terms of methods, it says the online version of the survey was distributed, but unclear to whom.

Also, the link to the survey question was distributed in the website, but not sent to patients? Who was selected for face-to-face interview? what criteria was there to select these participants? There is no response rate reported either. The methods section needs to be more clear. Maybe inclusion and exclusion criteria may help.

The main route through which the questionnaire was distributed was through the websites/social media of the patient-organizations mentioned in the text.  We changed the methods section to further clarify this point.

–Needs to define what was considered "chronic" and "acute" in terms of prescription use.

The definition of “chronic” and “acute” is explicitly given in the questionnaire.  We also added this to the methods section to clarify this point.

– The results section used the words, such as "majority" and "nearly always". Needs to report percentage for these.

Due to the nature of our sample we think that these descriptions were adequate but we added the exact numbers, as suggested, in the results section.

- In line 82, "50+ people" needs to be changed to "people with age of 50 or older"

This was changed in the text.

- Needs a description on "Personal Health Viewer"

A description was included. We explained this in the penultimate paragraph of the results section.

 - It is unclear how paperless electronic prescription can lead to online internet sale.

We explained this in the last paragraph of the results section by adding:” (by obviating a physical transfer of the prescription to the pharmacy)”

- In line 160-161, the abbreviation GPDR is used, but unclear where this came from.

GDPR is the abbreviation of the European Union initiative “General Data Protection Regulation”.  We clarified this in the text and added a reference to the European regulation.

Overall, the manuscript needs significant improvement. Also, the significance of study finding is questionable as the implementation of paperless electronic prescription is happening, regardless of this. The potential risk of this implementation is not described adequately 

We strongly disagree with this statement.  Why would the go-ahead in the implementation of paperless e-prescription in itself annihilate the significance of a study of the opinion of patients about these changes?  In addition, this study clearly points to significant problems resulting from the digital divide, the unresolved privacy issues and the infrastructure problems.  Reporting and documenting these problems will enable, we hope, accompanying initiatives to prevent vulnerable patients to be adversely affected by the implementation of paperless e-prescriptions.  That is the reason we, as scientists, point to the urgent need for a careful risk/benefit analysis of the proposed changes.

Reviewer 2 Report

This is a survey sent via email and face-to-face to several patient organizations in Belgium. The findings are interesting because this is the first study to query patient preferences to the receipt of electronic prescriptions instead of hand written.

The paper title might be changed to Patient perceptions of electronic prescriptions in Belgium: an exploratory policy analysis.

Among the recommendation that would strengthen the paper, the abstract could be formatted in using: Background:, Methods:, Results: Conclusion.The background should review the reasons why paperless systems have been adopted, primarily for safety and efficiency. Minimizing hand written prescriptions has been shown to reduce medication errors due to illegibility. In addition, the European Commission is likely to have additional documents supporting implementation of eprescribing that should be cited.Was the web page cited used as the basis to derive questions?

The methods section needs to be rewritten, and the authors should state whether the survey was reviewed in any way by an Institutional Review Board. There needs to be an estimate of the population that uses patient organizations to interact with health-related data and information in Belgium. There is no response rate calculated and stated, as is usual practice for surveys of this nature.Two different methods of survey information were used to collect data, and the results are combined when the results may be different between the groups. In addition, there is no breakdown among the patient organizations. Finally, there is no explanation of the type of statistics or systems used to generate percentages.

The results and discussion sections are well-written. In the results section, at lines 82-83, a citation is needed since the result was expected.

The conclusion needs to state what was derived from the study. As written, the paragraph at lines 190-198 is part of the discussion. It is significant that only 15% knew about the existence of the "Personal Health Viewer." The conclusion should state several important findings, especially if the finding was unexpected. In the abstract, the sentence at lines 23-26 is a good summation of findings, "Patients in these organizations may have inadequate knowledge and understanding of the implementation and impact of electronic prescription generation systems. Additional analyses of system risk and rewards from the patient perspective may be warranted."

Ethical Problem: You didn’t mention IRB review. In all likelihood, the study would be deemed as exempt by an IRB.

Author Response

This is a survey sent via email and face-to-face to several patient organizations in Belgium. The findings are interesting because this is the first study to query patient preferences to the receipt of electronic prescriptions instead of hand written.

We thank the reviewer for carefully reading the manuscript and for the many thoughtful and helpful suggestions.

The paper title might be changed to Patient perceptions of electronic prescriptions in Belgium: an exploratory policy analysis.

We accept this title change and thank the reviewer

Among the recommendation that would strengthen the paper, the abstract could be formatted in using: Background:, Methods:, Results: Conclusion.The background should review the reasons why paperless systems have been adopted, primarily for safety and efficiency. Minimizing hand written prescriptions has been shown to reduce medication errors due to illegibility. In addition, the European Commission is likely to have additional documents supporting implementation of eprescribing that should be cited.Was the web page cited used as the basis to derive questions?

Subdividing the abstract conflicts with the recommendations of the publisher.
We added references about safety and readability and about the European Union initiative in eHealth. (new reference 4 and 5)

The methods section needs to be rewritten, and the authors should state whether the survey was reviewed in any way by an Institutional Review Board. There needs to be an estimate of the population that uses patient organizations to interact with health-related data and information in Belgium. There is no response rate calculated and stated, as is usual practice for surveys of this nature.Two different methods of survey information were used to collect data, and the results are combined when the results may be different between the groups. In addition, there is no breakdown among the patient organizations. Finally, there is no explanation of the type of statistics or systems used to generate percentages.

We have made extensive changes in the methods section reflecting the suggestions of the reviewer.  Due to the exploratory nature of this research and the use of a convenience sample we did not carry out a very detailed statistical analysis as this would not be very informative. 

The results and discussion sections are well-written. In the results section, at lines 82-83, a citation is needed since the result was expected.

Citation is added (new reference 24)

The conclusion needs to state what was derived from the study. As written, the paragraph at lines 190-198 is part of the discussion.

This section was moved into the discussion

 It is significant that only 15% knew about the existence of the "Personal Health Viewer." The conclusion should state several important findings, especially if the finding was unexpected. In the abstract, the sentence at lines 23-26 is a good summation of findings, "Patients in these organizations may have inadequate knowledge and understanding of the implementation and impact of electronic prescription generation systems. Additional analyses of system risk and rewards from the patient perspective may be warranted."

We expanded the conclusion following these suggestions

Ethical Problem: You didn’t mention IRB review. In all likelihood, the study would be deemed as exempt by an IRB.

We added a statement to this effect

Round 2

Reviewer 1 Report

- line 34 - What are the potentially big benefits found in the references? Describing the findings from the references as "big benefits" is vague and unclear.

- line 37 - What are the possible risks and benefits of electronic prescribing? Since these 2 references have evaluated this already, summarizing these risks and benefits from the references may help readers to understand the purpose and objective of the author's study.

- line 42-43 - please replace "printed prescription" to another word. As you noted in your response and in the manuscript, these are not prescriptions, but rather a paper-based pointer.

- line 56-57 - how was e-prescription criticized? What were the main problems with e-prescription? Saying that the e-prescription was criticized without providing what the criticism was is not helping the readers.

- line 77 - Spell out "IRB" and which IRB approved this study as exampt?

- line 80 has "self-declared" and figure 1 has "self-reported"

- line 90 - provide percentage of total next to "clear majority"

- The authors responded saying that the definition of "chronic" use and "acute" use of prescription were added to methods, but I do not see them in method section. In medicine, "chronic" tends to be described as more than 30 days or more than 3 months. "Acute" can be less than 7 days or less than 14 days. The description used in the questionnaires are vague, thus it should be discussed in the method section that it was self-reported, patient individual opinion of chronic and acute.. 

- The method of this study is still questionable. It appears that the survey link was shared through website and social media, which makes it unclear how the sample was controlled. Was there duplicates? Can someone actually fill out the survey multiple times? There were 273 respondents, which appears to be enough to do statistical analysis or subgroup analysis. This open survey for anyone with computer access can potentially pose a serious flaw in the study design.  

Author Response

- line 34 - What are the potentially big benefits found in the references? Describing the findings from the references as "big benefits" is vague and unclear.

A sentence was added: “Big advantages are obvious in the speed, reliability, quantity, integration and analysis of information.”

- line 37 - What are the possible risks and benefits of electronic prescribing? Since these 2 references have evaluated this already, summarizing these risks and benefits from the references may help readers to understand the purpose and objective of the author's study.

These two papers were briefly situated:Possible benefits are a reduction of prescribing and medication errors, an increasein efficiencyand a limitation of fraud.  On the negative side, legal issues, privacy concerns, implementation costs and technical problems were noted.”

- line 42-43 - please replace "printed prescription" to another word. As you noted in your response and in the manuscript, these are not prescriptions, but rather a paper-based pointer.

We changed this to “printout” as suggested by the reviewer

- line 56-57 - how was e-prescription criticized? What were the main problems with e-prescription? Saying that the e-prescription was criticized without providing what the criticism was is not helping the readers.

This was added.  “….revealing practical problems, failed implementation attempt or the creation of additional barriers for vulnerable patients.”

- line 77 - Spell out "IRB" and which IRB approved this study as exampt?

We edited this sentence of the manuscript and spelled out the word IRB

- line 80 has "self-declared" and figure 1 has "self-reported"

We now use self-reported consistently in the manuscript

- line 90 - provide percentage of total next to "clear majority"

A percentage was added

- The authors responded saying that the definition of "chronic" use and "acute" use of prescription were added to methods, but I do not see them in method section. In medicine, "chronic" tends to be described as more than 30 days or more than 3 months. "Acute" can be less than 7 days or less than 14 days. The description used in the questionnaires are vague, thus it should be discussed in the method section that it was self-reported, patient individual opinion of chronic and acute.. 

The ”self-reported” nature of the word chronic was added in the method section

- The method of this study is still questionable. It appears that the survey link was shared through website and social media, which makes it unclear how the sample was controlled. Was there duplicates? Can someone actually fill out the survey multiple times? There were 273 respondents, which appears to be enough to do statistical analysis or subgroup analysis. This open survey for anyone with computer access can potentially pose a serious flaw in the study design.  

We already touch on the topic of the nature of our sample in the penultimate paragraph of the discussion and fully acknowledge this may be a limitation of the study.  We decided therefore that this would invalidate a more detailed statistical analysis of the data.  We do realize that more statistics are technically possible with our sample size but we disagree with the reviewer that such an analysis would be appropriate and informative. 

Reviewer 2 Report

Congratulations! A very interesting study that should assist with further policy development and implementation.

Author Response

Many thanks for the many insightful suggestions to improve our manuscript!